# Analysis and Validation of Sensitivity in Torque-Sensitive Actuators

**Minh Tran [1], Lukas Gabert [1,2] and Tommaso Lenzi [1,2,3,*]**

[1] Department of Mechanical Engineering, The Robotics Center, University of Utah, Salt Lake City, UT 84112, USA
[2] Rocky Mountain Center for Occupational and Environmental Health, Salt Lake City, UT 84111, USA
[3] Department of Biomedical Engineering, University of Utah, Salt Lake City, UT 84112, USA
**\*** Correspondence: t.lenzi@utah.edu

**Abstract:** Across different fields within robotics, there is a great need for lightweight, efficient actuators with human-like performance. Linkage-based passive variable transmissions and torque-sensitive transmissions have emerged as promising solutions to meet this need by significantly increasing actuator efficiency and power density, but their modeling and analysis remain an open research topic. In this paper, we introduce the sensitivity between input displacement and output torque as a key metric to analyze the performance of these complex mechanisms in dynamic tasks. We present the analytical model of sensitivity in the context of two different torque-sensitive transmission designs, and used this sensitivity metric to analyze the differences in their performance. Experiments with these designs implemented within a powered knee prosthesis were conducted, and results validated the sensitivity model as well as its role in predicting actuators' dynamic performance. Together with other design methods, sensitivity analysis is a valuable tool for designers to systematically analyze and create transmission systems capable of human-like physical behavior.

**Keywords:** torque-sensitive actuator; variable transmission; compliant actuator; robotics; prosthetics; legged locomotion





## 1. Introduction

Many robotic applications such as prosthetics, exoskeletons, humanoids, or collaborative robots require human-like physical behavior and performance [1–7]. To satisfy the demanding requirements of human-like behavior, robot actuators must provide a wide range of torque and speed at the output joint [8–11]. Due to their high efficiency and ability to be powered by portable batteries, electrical motors combined with custom geartrains are typically preferred to hydraulic, pneumatic, and other actuators [1]. However, the mechanical power output and electrical efficiency of electric motors drop sharply outside of a narrow torque–speed range [12]. Thus, it is difficult to satisfy the requirements for human-like physical behavior while simultaneously achieving high torque/power density and efficiency using a traditional actuation system with a fixed gear ratio.

To improve robot performance and reduce robot weight, researchers have proposed compliant actuation systems. Adding elastic elements in series [13–20], or parallel [21–23] to a geared DC motor can reduce the motor power demand during operation, improving electrical efficiency. Because the mechanical stiffness is constant, these compliant actuators must be tuned around a specific ambulation task or user to optimize efficiency [24]. Variable stiffness actuators address this issue by using a secondary motor to change the physical stiffness of the actuated joint, thus improving performance across tasks and users [25,26]. Polycentric and underactuated mechanisms have also demonstrated improved electrical efficiency and power density, but often at the cost of non-physiological joint kinematics and kinetics [27–32]. Mono- and bi-directional clutches save electrical energy by selectively disengaging the actuator from the joint output for specific portions of the task, or between

different tasks [33–35]. Antagonistic, multi-motor, and multi-joint actuator arrangements have also demonstrated energetic benefits [36–39]. However, these solutions also add considerable complexity and weight to the actuation system. Consequently, the power and torque density of robots using advanced compliant actuators is not necessarily better than that of robots using more conventional actuation systems. Thus, there is an unmet need for new actuators that can provide a wide range of torque and speed required for human-like physical behavior without increasing the robot's weight and size.

The analysis of human biomechanics reveals two common modes of operation across many joints: a fast-acting mode, where the joints move quickly to a desired position, and a slow-acting mode, where the joints must support and propel the body weight or exert large loads on external objects [8]. As the peaks of joint torque and speed are often not simultaneous, variable transmissions present an appealing solution to improve the actuator performance. Variable transmissions can improve electrical efficiency by adapting the transmission ratio based on the changing torque and speed demand at the output joint. Most variable transmissions are built to have an infinite range of motion (ROM) of the input and output joints [40], making them quite heavy and bulky. Actively variable transmissions with a limited ROM have been proposed to avoid this problem [41,42]. In an active variable transmission, a secondary motor changes the transmission ratio between different tasks (i.e., walking vs. stair ascent) by adjusting the actuation kinematics [41,42]. Actively variable transmissions are much lighter than continuous variable transmissions with an infinite range of motion. However, they can only change the transmission ratio slowly and when the actuator output torque is nearly zero, thus limiting the functional and theoretical improvements that can be achieved. These limitations can be overcome with passive counterparts that dynamically and continuously adjust the torque ratio in response to the output joint torque. Multiple passively variable transmissions and torque-sensitive transmissions have been developed specifically for robotic applications. They feature limited ROM and elastic features that react to output torques and adjust transmission properties. Prototypes have been validated on the bench [43,44], and also implemented in robotic fingers and hands [45,46]. Recently, both upper- and lower-limb prosthetic devices featuring these types of transmission have achieved unprecedented levels of power density and efficiency, increasing their clinical viability [47,48].

Although passively variable transmission and torque-sensitive transmission prototypes have demonstrated their promise, the modeling and analysis of these mechanisms remain an open research topic. Attempts to systematically analyze and design passively variable transmissions have mostly centered on the achievable range of transmission variation (e.g., dynamic range [34]), or on a quasi-static analysis of the transmission in specific ambulation tasks [36]. However, no analysis has been carried out regarding the dynamic interactions among the input joint, the output joint, and the elastic torque-sensitive element during operation. A lack of consideration of these dynamic interactions can lead to highly undesirable outcomes such as motor winding saturation [34,36], which can degrade robot performance or lead to controller instability. Relying solely on the intuition of the designer to develop mechanisms with effective and robust dynamic behaviors is challenging, as the relationships among the key variables are non-linear and can be highly complex.

In this paper, we introduce sensitivity (i.e., the ratio of change between an input variable and an output variable) as a new key metric to quantify and assess the capability of passively variable transmissions and torque-sensitive transmissions to perform dynamic tasks across a large range of motion. We formulate the mathematical definition of a key sensitivity variable and use this variable to analyze two different designs of a linkage-based torque-sensitive transmission system—a linear spring implementation (as introduced in [48]) and a torsional spring implementation—which are introduced for the first time in this paper. Hardware implementation of both these linear and torsional torque-sensitive transmissions within a powered knee prosthesis enables an experimental, quantitative comparison. The experimental results confirm the output of theoretical models showing the role of sensitivity in predicting the actuation performance. An experimentally validated

model for passively variable transmissions has the potential to enable lighter, more efficient, and more agile robots capable of human-like physical behavior.

## 2. Analytical Modeling

### 2.1. Kinematic Analysis

For both implementations, the kinematics of the proposed actuation system consist of two closed kinematic chains (Figure 1) acting in parallel. The first kinematic chain comprises five joints in closed configuration ($P_1R_2R_3P_2R_1$), creating a five-bar mechanism with two degrees of freedom. The second kinematic chain also comprises five joints in closed configuration ($R_4P_3R_3P_2R_1$) creating another five-bar mechanism with two degrees of freedom. In the proposed actuation system, $R_1$ is the output joint of the actuator, $P_1$ is the input joint of the actuator, and $P_2$ is a torque-sensitive joint whose position changes the relationship between the input joint $P_1$ and the output joint $R_1$. The position of the torque-sensitive joint $P_2$ is regulated by the second kinematic chain with a prismatic joint $P_3$, leading to a torque- and position-dependent torque ratio.

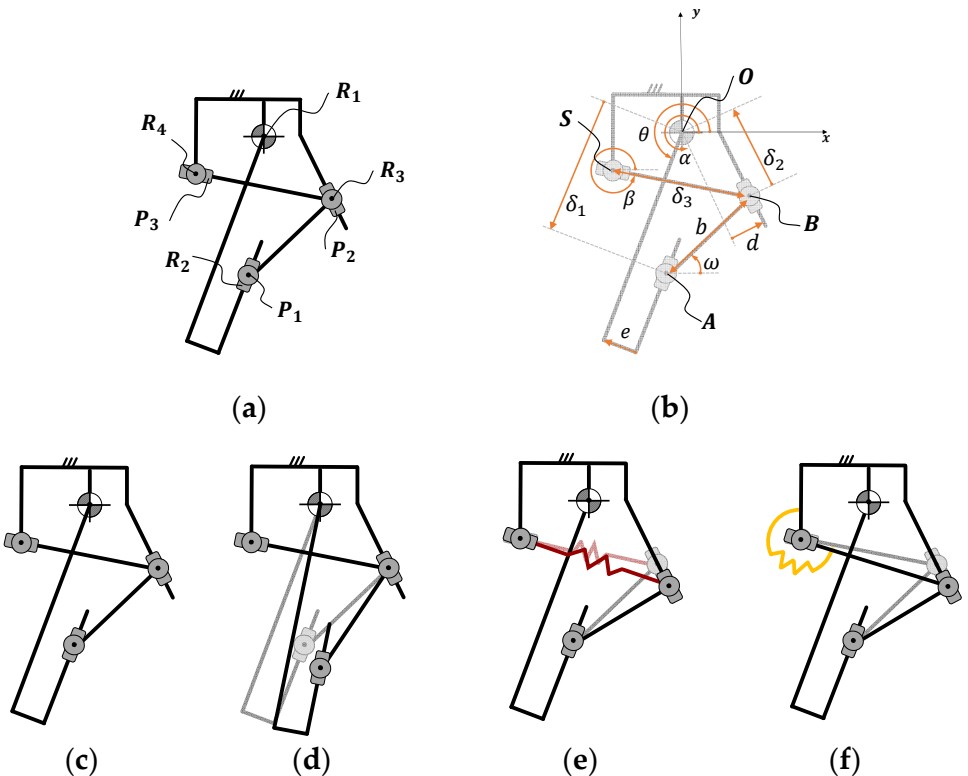

**Figure 1.** Kinematic diagram of the actuator with passive variable transmission. The proposed actuator is based on two closed kinematic chains acting in parallel (**a**,**b**). These parallel chains result in a system with two degrees of freedom: one is the rotation of the joint $R_1$ (**c**,**d**), and the other is the translation of the joint $P_3$. The position of joint $P_3$ can be regulated with either a linear spring (**e**) or a torsional spring (**f**).

To analyze the passive variable transmission, we first find the relationship of the position of the input joint ($\delta_1$) as a function of the position of the torque-sensitive joint ($\delta_2$). Then, we analyze the relationship between the two parallel kinematic chains to find the position of the torque-sensitive joint ($\delta_2$) as a function of the position of the passively actuated joint ($\delta_3$). In the following paragraphs, trigonometric functions $\sin(x)$, $\cos(x)$, and $\tan(x)$ are abbreviated as $s_x$, $c_x$, and $t_x$, respectively.

We start from the analysis for the first kinematic chain ($P_1R_2R_3P_2R_1$). For a given position of the torque-sensitive joint ($\delta_2$), we can obtain the relationship between the position of the input joint ($\delta_1$) and the position of the output joint ($\theta$) by imposing that the

$\overline{AB}$ distance equals the fixed link length (b), as shown in (1). Using the joint $R_1$ as the origin of the coordinate system (O), the position of B can be defined as a function of the position of the torque-sensitive joint $P_2$ ($\delta_2$) and its angle with respect to the x-axis ($\alpha$). Similarly, the position of A can be defined as a function of the position of the input joint $P_1$ ($\delta_1$) and the output joint ($\theta$).

$$\overline{AB} = \sqrt{(A_x - B_x)^2 + (A_y - B_y)^2} \equiv b \tag{1}$$

where $\begin{cases} B_x = \delta_2 c_\alpha + d c_{\alpha - \frac{\pi}{2}} \\ B_y = \delta_2 s_\alpha + d s_{\alpha - \frac{\pi}{2}} \end{cases}$ and $\begin{cases} A_x = \delta_1 c_\theta + e c_{\theta - \frac{\pi}{2}} \\ A_y = \delta_1 s_\theta + e s_{\theta - \frac{\pi}{2}} \end{cases}$

By solving (1) for $\delta_1$, we find the position of the input joint ($\delta_1$) as a function of the position of the output joint ($\theta$) and the position of the torque-sensitive joint ($\delta_2$).

$$\delta_1(\theta, \delta_2) = \sqrt{b^2 - (\delta_2 s_\alpha - e)^2 - d^2 c_{(\theta - \alpha)}^2 - 2 dec_{(\theta - \alpha)} - d\delta_2 s_{2(\theta - \alpha)}} + \delta_2 c_{(\theta - \alpha)} - d s_{(\theta - \alpha)} \tag{2}$$

Focusing on the second kinematic chain ($R_4 P_3 R_3 P_2 R_1$), we find the relationship between the position of the torque-sensitive joint ($\delta_2$) and the position of the passively actuated joint ($\delta_3$), as shown in (3).

$$\delta_2(\delta_3) = \sqrt{{\delta_3}^2 - (d - s_\alpha S_x + c_\alpha S_y)^2} + c_\alpha S_x + s_\alpha S_y \tag{3}$$

As we show in the next section, the main difference between the two implementations lies in the way that the joint $P_3$ adapts its position. In the linear spring implementation, a coil spring connecting joints $P_2$ and $P_3$ can extend and compress in reaction to the joint torque. In the torsional spring implementation, the reaction to joint torque is provided by a torsional spring located at the pivot $R_4$.

*2.2. Free-Body Diagram and Torque Ratio*

The torque ratio is an essential parameter of the passive variable transmission system, and is defined as the ratio between the resulting torque $\overrightarrow{T}$ on the output joint $R_1$ and the force $\overrightarrow{F_a}$ applied at the input joint $P_1$. Both the torque ratio and resulting joint torque are related to the joint angle and position of the torque-sensitive joint. To this end, we conducted a free-body diagram analysis, which we present separately for the linear and torsional spring implementations.

Starting from the kinematic model, we perform a free-body diagram analysis (Figure 2) to determine the torque ratio ($\tau_{torque}$) of the linear implementation. To this end, we first find the torque ratio ($\tau_{torque}$) as a function of the position of the passively actuated joint ($\delta_3$) and the output joint ($\theta$). Then, we model the relationship between the position of the passively actuated joint ($\delta_3$) and the output torque ($\overrightarrow{T}$) as a function of the stiffness (k), rest length ($\delta_0$), and preload ($F_{s0}$) of the spring. Finally, we combine these two relationships to relate the torque ratio ($\tau_{torque}$) to the output position ($\theta$) and torque ($\overrightarrow{T}$). In the following analysis, we define action ($\overrightarrow{F_{xy}}$) and reaction forces ($\overrightarrow{R_{xy}}$) so that, $\overrightarrow{F_{xy}} = -\overrightarrow{R_{xy}}$.

Focusing on the input joint ($P_1$), we see that the input force $\overrightarrow{F_a}$ is in line with the prismatic joint $P_1$ (Figure 2b). This input force is balanced by a reaction force ($\overrightarrow{R_{P1}}$) perpendicular to the direction of the prismatic joint $P_1$ and a reaction force ($\overrightarrow{R_b}$) in line with the connecting bar, leading to the force equilibrium shown in (4).

$$\overrightarrow{F_a} + \overrightarrow{R_{P1}} + \overrightarrow{R_b} = \overrightarrow{F_a} + \overrightarrow{R_{P1}} - \overrightarrow{F_b} = \overrightarrow{0} \tag{4}$$

The connecting bar acts as a two-force body due to the revolute joints $R_2$ and $R_3$ (Figure 2c). Thus, the action force of the connecting bar ($\overrightarrow{F_b}$) depends on the relative orientation between the bar segments b and $P_1$, as shown in (5).

$$\begin{cases} |F_a| = -|R_b|c_{(\omega+\pi-\theta)} = |F_b|c_{(\omega+\pi-\theta)} \\ |R_{P1}| = -|R_b|s_{(\omega+\pi-\theta)} = |F_b|s_{(\omega+\pi-\theta)} \end{cases} \tag{5}$$

where $\omega$ is the angle of the connecting bar with respect to the reference frame xy (Figure 2b), which is defined in (6).

$$\omega = \text{atan2}\left(\overline{OB}s_{\widehat{OB}} - \delta_1 s_\theta + ec_\theta \; \overline{OB}s_{\widehat{OB}} - \delta_1 s_\theta + ec_\theta\right) \tag{6}$$

where $\begin{cases} \overline{OB} = \sqrt{B_x^2 + B_y^2} \\ \widehat{OB} = \text{atan2}(B_y, B_x) \end{cases}$

The action of the connecting bar $(\vec{F_b})$ is balanced by a reaction force $(\vec{R_{P2}})$ perpendicular to the prismatic joint $P_2$ as well as the reaction force $(\vec{R_s})$ generated by the tension spring (Figure 2d). Moreover, if the prismatic joint $P_2$ contacts the mechanical end-stop at the end of its range of motion, a reaction force $(\vec{R_{ES}})$, longitudinal to the prismatic joint $P_2$, is also present. This force balance is shown in (7) and in Figure 2d.

$$\vec{R_s} + \vec{R_{ES}} + \vec{R_{P2}} + \vec{F_b} = \vec{0} \rightarrow \vec{F_b} = \vec{F_s} + \vec{F_{ES}} + \vec{F_{P2}} \tag{7}$$

The forces generated by the spring $(\vec{F_S})$, the prismatic joint $P_2$ $(\vec{F_{P2}})$, and the mechanical end-stops $(\vec{F_{ES}})$ are balanced by a torque at the output joint (O) (Figure 2e). Because $\vec{F_s}$ is in line with $\overline{BS}$ (Figure 2f), the force balance can be simplified, as shown in (8).

$$\vec{T} + \vec{OB} \times \left(\vec{F_{Es}} + \vec{F_{P2}} + \vec{F_s}\right) = \vec{0} \tag{8}$$

By substituting (7) and (8), we find (9) relating the connecting bar force $(\vec{F_b})$ to the output torque $(\vec{T})$.

$$\vec{T} + \vec{OB} \times \vec{F_b} = \vec{0} \rightarrow \frac{|T|}{|F_b|} = -\overline{OB}\left(c_{\widehat{OB}}s_\omega - s_{\widehat{OB}}c_\omega\right) \tag{9}$$

By combining (7) and (8), we find $|F_b|$ as a function of $|F_a|$. Then, by plugging this relationship in (9), we obtain the ratio between the output torque (T) and the input force ($F_a$) (i.e., the torque ratio ($\tau_{\text{torque}}$)), as shown in (10).

$$\tau_{\text{torque}}\left(\theta, \widehat{OB}, \overline{OB}\right) = \frac{|T|}{|F_a|} = \overline{OB}s_{\theta-\widehat{OB}} + \frac{\left(\overline{OB}c_{\theta-\widehat{OB}}\right)\left(\overline{OB}s_{\theta-\widehat{OB}} - e\right)}{\sqrt{b^2 - \left(\overline{OB}s_{\theta-\widehat{OB}} - e\right)^2}} \tag{10}$$

It is important to note that the angle $\widehat{OB}$ and the distance $\overline{OB}$ depend directly on the position of the torque-sensitive joint $\delta_2$, and, consequently, on the passively actuated joint $\delta_3$, as shown in (11). The torque ratio thus depends on the position of the output joint $\theta$ and the passively actuated joint ($\delta_3$).

$$\begin{cases} \overline{OB} = \sqrt{\begin{array}{c} \left(c_\alpha\left(\sqrt{\delta_3^2 - (d - s_\alpha S_x + c_\alpha S_y)^2} + c_\alpha S_x + s_\alpha S_y\right) + ds_\alpha\right)^2 + \\ \left(s_\alpha\left(\sqrt{\delta_3^2 - (d - s_\alpha S_x + c_\alpha S_y)^2} + c_\alpha S_x + s_\alpha S_y\right) - dc_\alpha\right)^2 \end{array}} \\ \widehat{OB} = \text{atan2}\left(\begin{array}{c} s_\alpha\left(\sqrt{\delta_3^2 - (d - s_\alpha S_x + c_\alpha S_y)^2} + c_\alpha S_x + s_\alpha S_y\right) - dc_\alpha, \\ c_\alpha\left(\sqrt{\delta_3^2 - (d - s_\alpha S_x + c_\alpha S_y)^2} + c_\alpha S_x + s_\alpha S_y\right) + ds_\alpha \end{array}\right) \end{cases} \tag{11}$$

By combining (10) and (11), we obtain the torque ratio as a function of the position of the passively actuated joint ($\delta_3$).

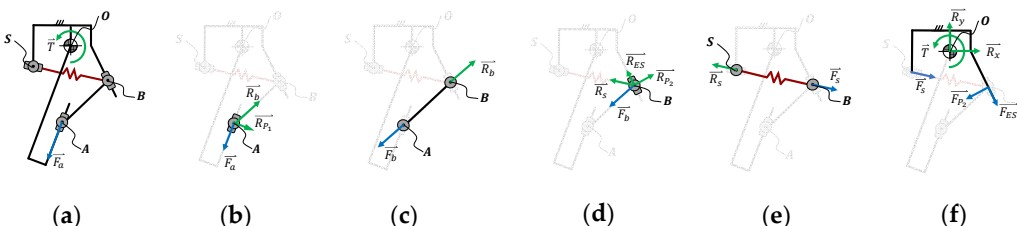

**Figure 2.** The torque ratio (torque at the output (T) divided by the force at the input ($F_a$)) of the linear implementation is found by performing a free-body diagram analysis of the proposed mechanism (**a**), including the input slider joint (**b**), the connecting bar (**c**), the tuning joint (**d**), the passively actuated joint (**e**), and the output joint (**f**).

The free-body diagram analysis (Figure 2d) shows that the position of the passively actuated joint ($\delta_3$) depends on the force equilibrium. In other terms, the position of the passively actuated joint ($\delta_3$) depends on the force acting on the spring actuating it ($F_s$), which, in turn, depends on the output torque. Thus, the torque ratio depends on the output torque. To solve this force equilibrium, we need to find the relationship between the spring force ($\vec{F_s}$) and the position of the passively actuated joint ($\delta_3$). As shown in Figure 2e, the spring force ($\vec{F_s}$) is in line with the prismatic joint $P_3$ and depends on the stiffness (k), the rest length ($\delta_0$), and the preload ($F_{s0}$), as shown in (12).

$$|F_s| = k\,(\delta_3 - \delta_0) + F_{s0} \tag{12}$$

Combining (12) and (7), we obtain the force balance shown in (13, 14), which defines the forces perpendicular ($F_{P2}$) and longitudinal ($F_{ES}$) to the prismatic joint $P_2$ as a function of $\delta_3$ and $F_b$.

$$\begin{bmatrix} c_\alpha & s_\alpha \\ s_\alpha & -c_\alpha \end{bmatrix} \begin{bmatrix} |F_{P2}| \\ |F_{Es}| \end{bmatrix} = \begin{bmatrix} -(k\,(\delta_3-\delta_0)+F_{s0})c_\beta - |F_b|c_\omega \\ -(k(\delta_3-\delta_0)+F_{s0})s_\beta - |F_b|s_\omega \end{bmatrix} \tag{13}$$

$$\begin{cases} F_{ES} = -((k(\delta_3-\delta_0)+F_{s0})c_\beta + F_b c_\omega)c_\alpha \\ \qquad -((k(\delta_3-\delta_0)+F_{s0})s_\beta + F_b s_\omega)s_\alpha \\ F_{P2} = -((k(\delta_3-\delta_0)+F_{s0})c_\beta + F_b c_\omega)s_\alpha \\ \qquad +((k(\delta_3-\delta_0)+F_{s0})s_\beta + F_b s_\omega)c_\alpha \end{cases} \tag{14}$$

When the torque-sensitive joint $\delta_3$ does not rest on the mechanical end-stops, the longitudinal force $F_{ES} = 0$. We combine (9) and (14) to find the output torque as a function of $\delta_3$ and $\theta$ and their derived variables, as shown in (17). Finally, we can then find how the torque ratio changes as a function of the output torque.

$$|T(\delta_3)| = \left( \frac{(c_\beta c_\alpha + s_\beta s_\alpha)}{(c_\alpha c_\omega + s_\alpha s_\omega)} (k(\delta_3 - \delta_0) + F_{s0}) \right) \overline{OB}(c_{\widehat{OB}}s_\omega - s_{\widehat{OB}}c_\omega) \tag{15}$$

Because the position of the torque-sensitive joint ($\delta_2$) depends on the position of the passively actuated joint ($\delta_3$), the torque ratio ($\tau_{\text{torque}}$) changes as a function of the positions of both the output joint ($\theta$) and the passively actuated joint ($\delta_3$). As shown in (12), the position of the passively actuated joint ($\delta_3$) depends on the spring force ($F_s$), which, in turn, depends on the torque on the output joint (T). As a result, the torque ratio depends on the output torque. When the actuation system is under a certain load (i.e., input force and output torque are non-zero), the spring in the passively actuated joint ($P_3$) extends. The spring extension causes the torque-sensitive joint ($P_2$) to move, increasing the moment arm

of the force acting on the connecting bar ($F_B$) with respect to the output joint (O). Thus, under certain conditions, the torque ratio increases with the torque on the output joint.

The speed ratio $\tau_{speed}$ between the speed of the input and output joints depends on the position and velocity of the output joint $\theta$ and the passively actuated joint $\delta_3$. Due to the presence of an elastic element, the torque ratio and the speed ratio are not inversely correlated as with fixed transmission systems.

As the linear and torsional spring implementations share similar kinematic structures, the equation of the torque ratio $\tau_{torque}$, as shown in Equations (9) and (10), also applies to the torsional implementation. However, there is a difference in the relationship of the output torque with respect to the output angle and the extension of the elastic torque-sensitive element.

The free-body diagram analysis (Figure 3d,e) shows that the position of the passively actuated joint ($\delta_3$) and, consequently, the rotation of the torsional spring pivot ($\beta$) depends on the force equilibrium. In other words, $\beta$ depends on the torque acting of the torsional spring ($T_s$), which, in turn, depends on the output torque. To solve the force equilibrium, we need to first find the relationship between the spring torque ($\overrightarrow{T_s}$) and the position of the spring pivot ($\beta$). As shown in Figure 3e, the spring torque ($\overrightarrow{T_s}$) coincides with the revolute joint $R_4$ and depends on the torsional stiffness ($k_{tor}$), the rest angle of the spring ($\beta_0$), and the preload ($T_{s0}$), as shown in (16).

$$|T_s| = k_{tor}\,(\beta - \beta_0) + T_{s0} \tag{16}$$

This spring torque results from the force $F_s$, as shown in Equation (17).

$$|F_s| = \frac{T_s}{\delta_3} \tag{17}$$

Combining (16) and (7), we obtain the force balance shown in (18), which defines the forces that are perpendicular ($F_{P2}$) and longitudinal ($F_{ES}$) to the prismatic joint $P_2$ as a function of $\delta_3$ and $F_b$.

$$\begin{bmatrix} c_\alpha & s_\alpha \\ s_\alpha & -c_\alpha \end{bmatrix} \begin{bmatrix} |F_{P2}| \\ |F_{Es}| \end{bmatrix} = \begin{bmatrix} -\frac{(k_{tor}\,(\beta-\beta_0)+T_{s0})}{\delta_3}s_\beta - |F_b|c_\omega \\ \frac{(k_{tor}\,(\beta-\beta_0)+T_{s0})}{\delta_3}c_\beta - |F_b|s_\omega \end{bmatrix} \tag{18}$$

$$\begin{cases} F_{ES} = -\left(\left(\frac{(k_{tor}\,(\beta-\beta_0)+T_{s0})}{\delta_3}\right)s_\beta + F_b c_\omega\right)c_\alpha \\ \quad -\left(-\left(\frac{(k_{tor}\,(\beta-\beta_0)+T_{s0})}{\delta_3}\right)c_\beta + F_b s_\omega\right)s_\alpha \\ F_{P2} = -\left(\left(\frac{(k_{tor}\,(\beta-\beta_0)+T_{s0})}{\delta_3}\right)s_\beta + F_b c_\omega\right)s_\alpha \\ \quad +\left(-\left(\frac{(k_{tor}\,(\beta-\beta_0)+T_{s0})}{\delta_3}\right)c_\beta + F_b s_\omega\right)c_\alpha \end{cases} \tag{19}$$

When the torque-sensitive joint $\delta_3$ does not rest on the mechanical end-stops, the longitudinal force $F_{ES} = 0$. We combine (7) and (19) to find the output torque (T) as a function of $\delta_3$ and $\theta$ and their derived variables, as shown in (15). Finally, we can then find how the torque ratio changes as a function of the output torque.

$$|T(\delta_3)| = \left(\frac{(s_\alpha c_\beta - c_\alpha s_\beta)}{(c_\alpha c_\omega + s_\alpha s_\omega)}\left(\frac{(k_{tor}\,(\beta - \beta_0) + T_{s0})}{\delta_3}\right)\right)\overline{OB}(c_{\widehat{OB}}s_\omega - s_{\widehat{OB}}c_\omega) \tag{20}$$

It can be seen from (20) and (15) that, for the torsional implementation, the torque ratio ($\tau_{torque}$) and the output torque (T) are both functions of the positions of the output joint ($\theta$) and the passively actuated joint ($\delta_3$). Consequently, similar to the linear implementation, the torque ratio ($\tau_{torque}$) is a function of the output joint ($\theta$) and output torque (T).

Our torque-sensitive actuator is intended to be driven by a DC motor through a primary gear transmission and a ballscrew, which is a common configuration for linkage-

based robotic joints [48–50]. The total torque ratio between the output joint and the motor takes into account the gear and ballscrew stages, as shown in Equation (21).

$$\tau_{torque,total} = \tau_{torque} \times \tau_{gear} \times \tau_{screw} \tag{21}$$

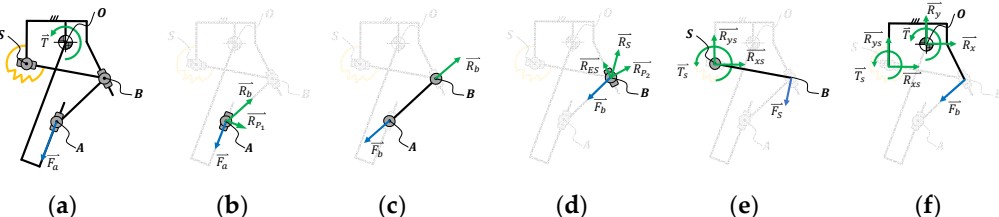

(**a**)        (**b**)        (**c**)        (**d**)        (**e**)        (**f**)

**Figure 3.** The torque ratio (torque at the output (T) divided by the force at the input ($F_a$ )) of the torsional implementation is found by performing a free-body diagram analysis of the proposed mechanism (**a**), including the input slider joint (**b**), the connecting bar (**c**), the tuning joint (**d**), the passively actuated joint (**e**), and the output joint (**f**).

## 3. Formulation of Sensitivity

Sensitivity, defined as the impact of an infinitesimal parameter change on a behavior of interest, is a useful metric for the analysis of dynamic systems [51]. For the analysis of passive variable transmission systems, it is useful to quantify the sensitivity of the displacement of the input ($\delta_1$) to a change in torque at the output joint (T). To this end, we propose using a local and first-order sensitivity $S_T^{\delta_1}\big|_\theta$ which is the ratio of the change in input position ($\delta_1$) with respect to a change in output torque (T) under a certain output angle ($\theta$), as shown in Equation (24).

$$S_T^{\delta_1}\bigg|_\theta = \frac{\Delta\delta_1(\theta,\delta_3)}{\Delta T(\theta,\delta_3)}\bigg|_\theta \tag{22}$$

The proposed sensitivity ($S_T^{\delta_1}\big|_\theta$) quantifies the displacement of the prismatic input joint ($\delta_1$) (i.e., the movement of the linear actuator) necessary to obtain a desired change in output torque. The higher the sensitivity, the more the linear actuator must move to generate a desired output torque. In other terms, a high sensitivity indicates that the linear actuator needs to perform a large movement and a low sensitivity indicates that the linear actuator needs to perform a small movement. Therefore, increasing the sensitivity increases the speed requirement on the linear actuator, which may lead to voltage saturation and degradation of control performance. On the other hand, decreasing the sensitivity may result in a system that does not sufficiently increase the torque ratio with the output torque, which can lead to over-heating of the motor and low efficiency. Thus, sensitivity is a useful parameter for modeling and design of torque-sensitive actuators, especially to compare the performance of different actuator designs and configurations.

## 4. Mechatronic Implementation

To experimentally verify the analytical model of the torque-sensitive actuator, we implemented two passive variable transmission systems using a linear spring and a torsional spring to passively actuate the torque-sensitive joint. The specific design parameters for the two actuators are shown in Tables 1 and 2. Both implementations of the torque-sensitive actuator are implemented in an autonomous powered knee prosthesis prototype previously presented [48] (Figure 4a). Briefly, both implementations are actuated by the same linear actuator and control electronics (Figure 4b). The linear actuator comprises a brushless DC motor (Maxon Motor EC-4pole 22, 24 V, 120 W) that drives a ballscrew (Ewellix, pitch diameter 12 mm, lead 2 mm) through a helical gear pair (Boston Gears, 24 DP, 12:30 gear ratio). The control electronic comprises two microcontrollers (PIC32) to process data from sensors and run control algorithms; a power electronics board with a motor current driver

(Elmo Gold Twitter) and motor chokes; and a motherboard with an embedded computer (Raspberry Pi 3+ compute module) to save data and communicate via Wi-Fi to an external laptop for data telemetry. An embedded 8-cell 850 mAh lithium-polymer battery powers the device. A detailed description of the mechatronic implementation is provided in our previous publication [48].

**Table 1.** Parameters for Linear Implementation.

| Symbol | Value |
|:---:|:---:|
| $\delta_2$, l | 22 mm |
| $\delta_2$, h | 35 mm |
| $F_{s0}$ | 150 N |
| K | 35 N/mm |
| $S_x$ | −5 mm |
| $S_y$ | −22 mm |
| b | 107 mm |
| e | 35 mm |
| d | −16 mm |
| $\alpha$ | 275° |
| Lead | 2 mm |
| Gear ratio | 2.5 |

**Table 2.** Parameters for Torsional Implementation.

| Symbol | Value |
|:---:|:---:|
| $\delta_2, l$ | 22 mm |
| $\delta_2, h$ | 35 mm |
| $T_{s0}$ | 5 Nm |
| $k_{tor}$ | 1.1 Nm/° |
| $S_x$ | −10 mm |
| $S_y$ | −33 mm |
| b | 107 mm |
| e | 35 mm |
| d | −19 mm |
| $\alpha$ | 272° |
| Lead | 2 mm |
| Gear ratio | 2.5 |

In both implementations, the linear actuator is connected to the torque-sensitive actuator through two connecting bars, which are joined by a steel shaft. The steel shaft slides into two parallel slots in the top knee structure and connects to an elastic element. In the linear implementation (Figure 4c), the elastic element is realized with a tension coil spring (Century Spring 6056CS). A magnetic rotary encoder (RLS RM08, 12-bit incremental) is located at the base pivot of the spring and measures the movement of the torque-sensitive joint. In the torsional implementation (Figure 4d), the elastic element is realized with two custom torsion springs (music wire, 4 mm wire diameter, 1 3/8 active coils) that connect to the connecting bar shaft through a linear bearing (Misumi LM5ML). A linear transducer (P3 America LMC8, 8 mm stroke/10 k ohms) measures the movement of the torque-sensitive joint. The designs of the linear and torsional implementations resulted in similar levels of torque ratio variation as well as the bounding profiles of torque ratio (Figure 4e).

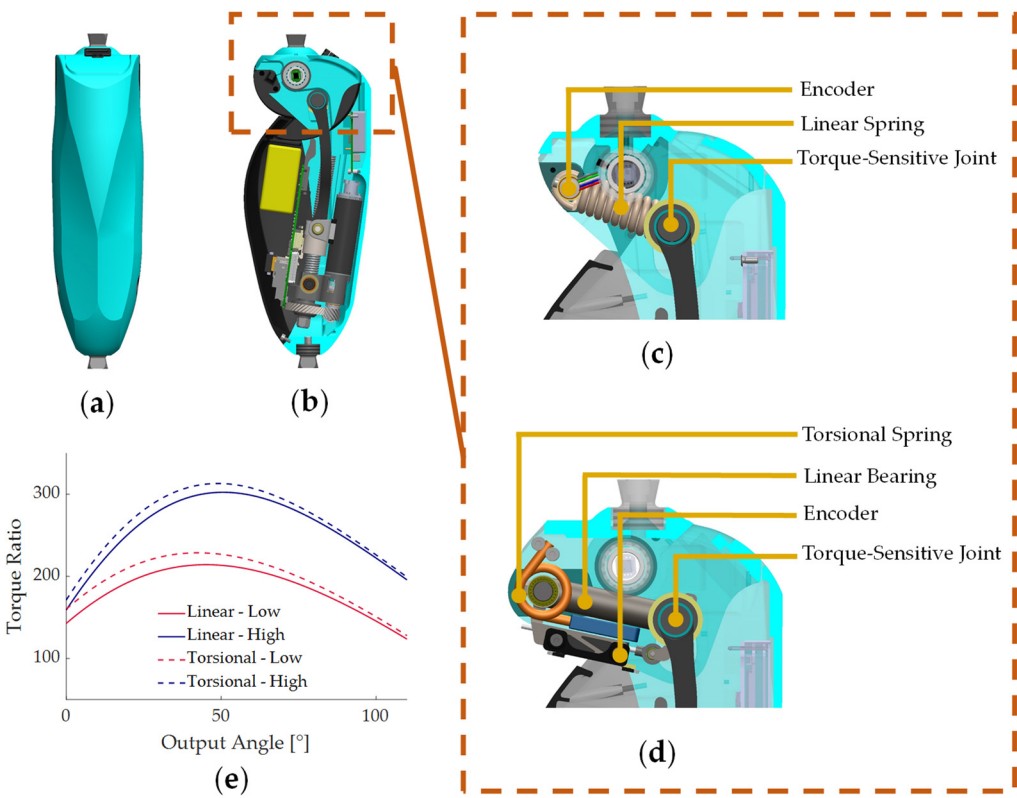

**Figure 4.** Mechatronic Implementation of the knee prosthesis used as a testing apparatus for this study. (**a**) Front view of the knee prosthesis. (**b**) Cross section of prosthesis side view, showing main electronic and primary transmission components. (**c**) Close-up view of the linear implementation of the torque-sensitive system. (**d**) Close-up view of the torsional implementation of the torque-sensitive system. (**e**) Range of torque ratio variation of the linear and torsional implementations.

## 5. Sensitivity Analysis

We performed a sensitivity analysis to compare the dynamic performance of the two actuator implementations. To obtain a comprehensive assessment, we compared the sensitivity across the range of motions of the output angle ($\theta$) and the torque-sensitive joint movement ($\delta_3$). First, we divided the range of motions of the output angle ($\theta$) and the torque-sensitive joint displacement ($\delta_3$) into 500 equally spaced values each, obtaining a $500 \times 500$ grid of the operating space. Then, for each point in this $500 \times 500$ grid, we calculated the values of the input position $\delta_1$ and of the output torque T (Figure 5a–d). For both the input position $\delta_1$ and the output torque T, we calculated the change from each node of the operating space to the node directly next to it. This operation resulted in two difference matrices, $\Delta\delta_1$ and $\Delta$T. To obtain the sensitivity matrix $S_T^{\delta_1}\big|_\theta$, we divided the input difference matrix $\Delta\delta_1$ by the output difference matrix $\Delta$T (Figure 5e,f). Finally, we calculated a relative sensitivity matrix by taking the element-by-element division of the sensitivity matrix of the torsional implementation by the sensitivity matrix of the linear implementation (Figure 5g).

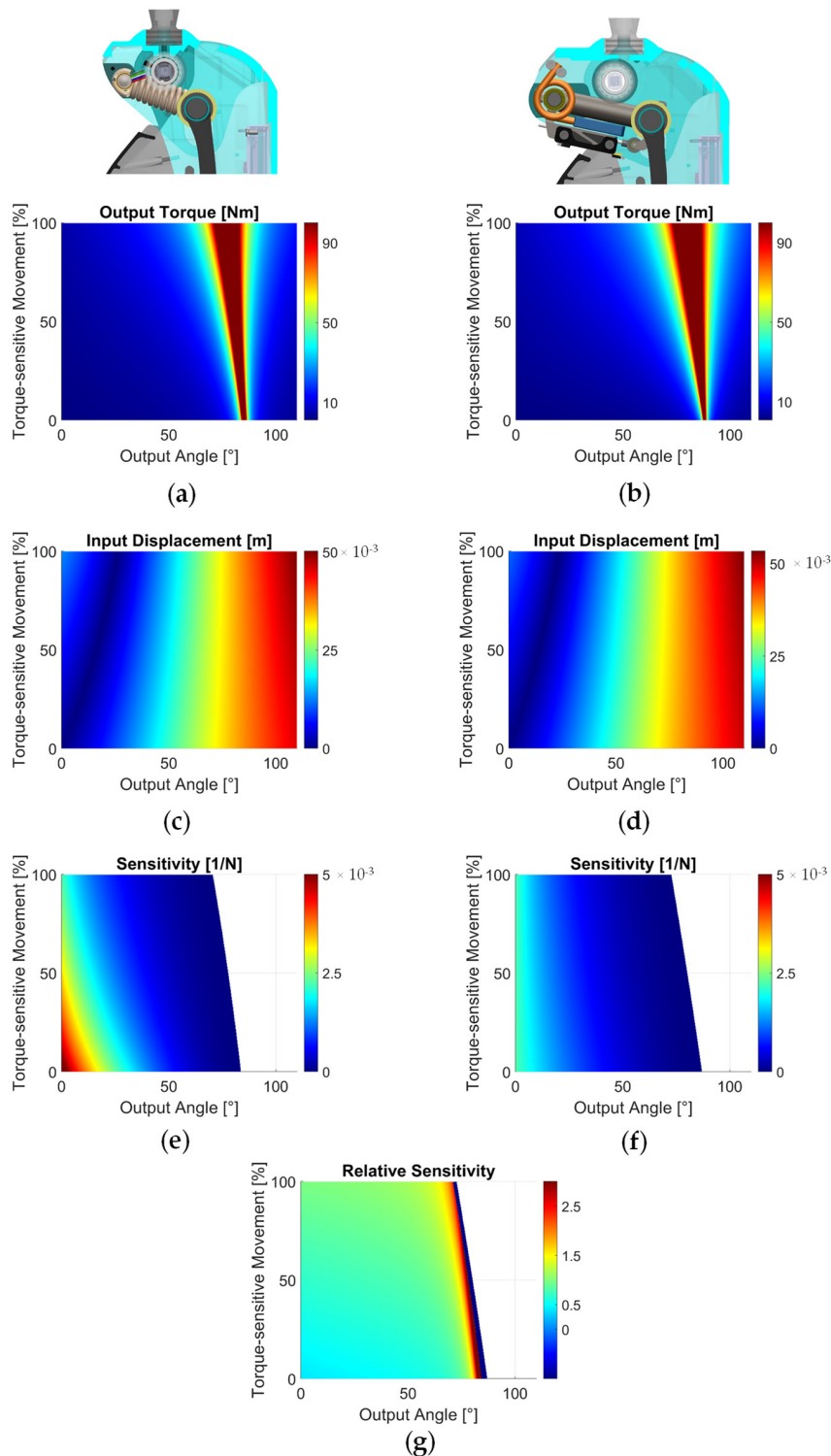

**Figure 5.** Calculated output torque and input displacement across the operation space. (**a**) Output torque heatmap of the linear implementation. (**b**) Output torque heatmap of the torsional implementation. The value of output torque is capped at 100 Nm to allow for a meaningful visualization across the operation space. (**c**) Input displacement heatmap of the linear implementation. (**d**) Input displacement heatmap of the torsional implementation. Displacements shown in Figure 5**c**,**d** are relative values with respect to the neutral position (i.e., 0° output angle and 0% spring extension) of each implementation. **Calculated sensitivity across the operation space.** (**e**) Sensitivity heatmap of the linear implementation. (**f**) Sensitivity heatmap of the torsional implementation. (**g**) Relative sensitivity of the torsional implementation with respect to the linear implementation.

The input position and output torque profiles show similar magnitudes and trends across the two implementations (Figure 5a–d). In particular, the output torque increased non-linearly with both the output angle and the extension of the torque-sensitive spring (Figure 5a,b). The input displacement increases as the output angle increases, and decreases as the output angle increases (Figure 5c,d). However, the sensitivity profiles show a significant difference between the two implementations. For the linear implementation, sensitivity is highest near the neutral position ($\sim 5.5 \times 10^{-3}$ N$^{-1}$), and rapidly decreases with increasing output angle and spring movement (Figure 5e). For the torsional implementation, sensitivity near the neutral position is about half that of the linear implementation ($\sim 2.3 \times 10^{-3}$ N$^{-1}$), but decreases more gradually through the operation space. There are regions where sensitivity is not defined and, for these regions, the torque-sensitive joint cannot extend even when 100 Nm or more of output torque is applied. The heatmap of relative sensitivity between the two implementations (Figure 5g) shows that the torsional implementation is less sensitive than the linear implementation at lower output angles and torque-sensitive spring extension (relative sensitivity $\sim 0.5$–1), but much more sensitive than the linear implementation at the regions of higher output angle (relative sensitivity >2.5).

## 6. Experiments

### 6.1. Joint Torque and Sensitivity Characterization

In this paper, we chose to use an open-loop torque controller, meaning that the commanded joint torque is converted to a commanded motor current through a model of the transmission system. This open-loop strategy is very common in powered prosthetics, as it eliminates the need for torque-sensing components and simplifies the system hardware. To quantify the accuracy of the open-loop torque controller and sensitivity models, we commanded a torque ramp to the knee prostheses, which was fixed at an output angle of 10° by a custom testing jig (Figure 6a). The torque profile was determined so that the torque-sensitive joint can travel through its full range of motion. For each implementation (i.e., linear, torsional), we performed five repetitions. As the commanded torque gradually ramped up to the desired level, the input slider movement and output torque were measured with a 6-axis loadcell (Sunrise Instrument M3713D) and on-board encoders. The modeled sensitivity was obtained with the output angle and torque-sensitive joint displacement $\delta_2$ similar to Figure 5e,f. The measured sensitivity was obtained by direct calculation using Equation (24) with the measured output torque and input slider displacement, shown in Figure 6b,c.

Figure 6b shows the commanded and measured torques for both implementations. The average torque-tracking error was $0.39 \pm 0.13$ and $0.36 \pm 0.15$ Nm for the linear and torsional implementations, respectively. The final torque-tracking error at the end of the torque ramp command was $0.19 \pm 0.16$ and $0.33 \pm 0.22$ Nm for the linear and torsional implementations, respectively. Figure 6d shows the modeled and measured sensitivity profiles during the commanded torque ramp. Throughout the movement of the torque-sensitive joint $\delta_2$ from its minimum to maximum positions, the trend and magnitude of the measured sensitivity profiles closely matched the modeled values for both implementations. In particular, the average error between the modeled and measured sensitivity through the test was $0.51 \pm 0.29$ and $0.32 \pm 0.24$ N$^{-1}$ for the linear and torsional implementations, respectively.

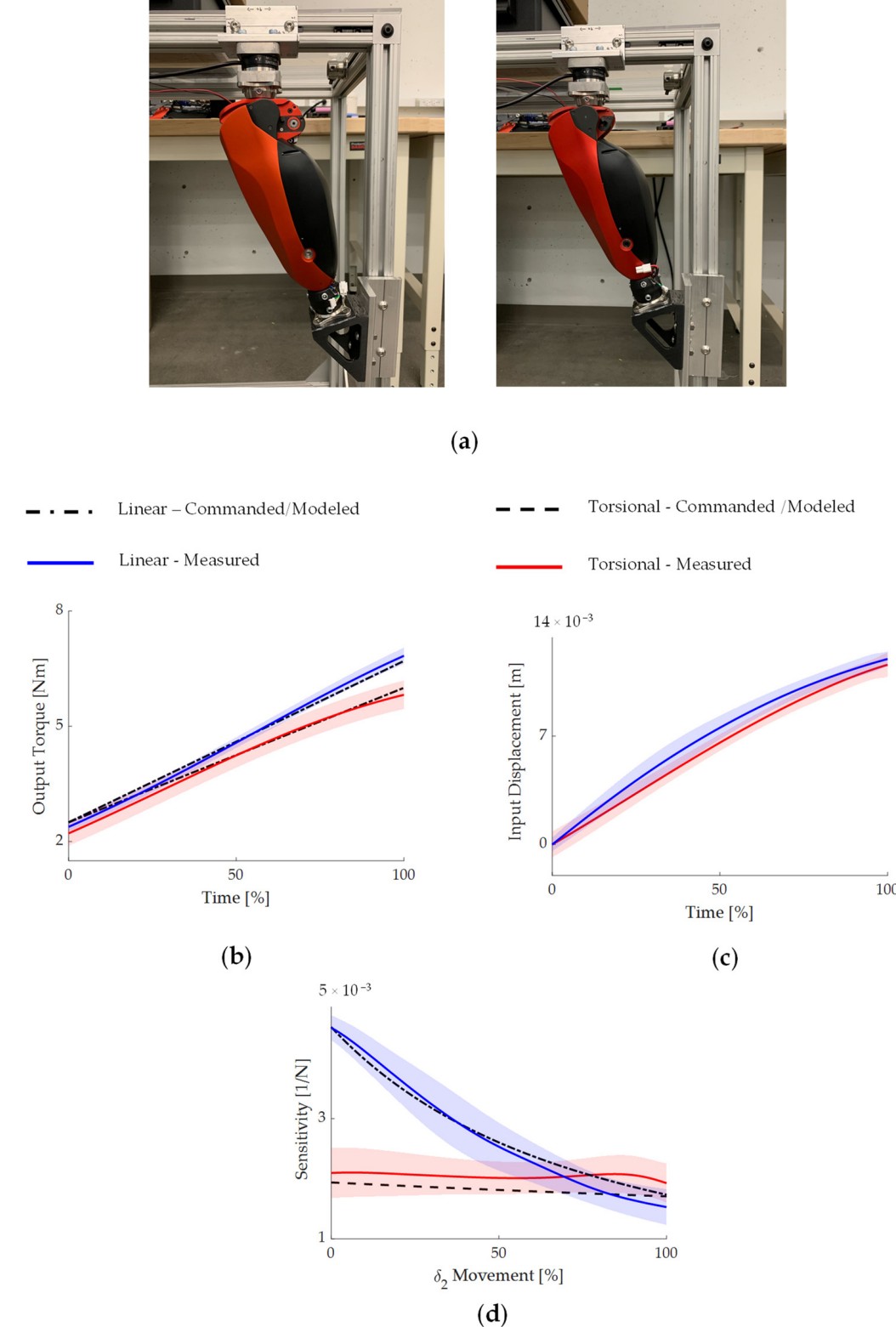

**Figure 6.** Output torque tracking and sensitivity quantification on the bench. (**a**) Experimental setup. The knee joint was fixed in a testing rig, and a 6-axis loadcell measures the output torque. (**b**) Commanded and measured output torque. (**c**) Measured input displacement. (**d**) Modeled and measured sensitivity. Results are shown for both implementations.

## 6.2. Damping Emulation

To evaluate the ability of each implementation to provide torque in dynamic tasks at lower output angles, we performed backdriving tests in which the knee prosthesis prototype was programmed to emulate a damper (providing resistive torque proportional to output velocity). Damping emulation is a crucial function for robotic joints; in particular, knee prostheses during walking and stair descent [48]. For this test, we commanded three levels of damping coefficient at the output joint (0.1, 0.2, 0.3 Nms/°). For each level of damping coefficient, we fixed the shank portion to a bench, and an experimenter manually moved the thigh portion through a similar sinusoidal motion between 5 and 30° output angle at 0.5 Hz (as synchronized with a metronome signal) for 7 s (Figure 7a). We measured the output torque with a 6-axis loadcell (Sunrise Instrument M3713D). Figure 7b shows the commanded and measured torques as well as the movement of the torque-sensitive joint under different desired levels of damping for both implementations. For the damping level of 0.1 Nms/°, both implementations were stable, and the average errors between the commanded and measured torques were $0.81 \pm 1.12$ Nm and $0.51 \pm 0.41$ Nm for the linear and torsional implementations, respectively. It is worth noting that the profiles of the torque-sensitive joint $\delta_2$ were different between the two implementations, likely due to their different sensitivity characteristics. For the damping level of 0.2 Nms/°, the linear implementation experienced marginal instability. The sudden and unusual movement of the torque-sensitive joint $\delta_2$ and the large error in torque tracking indicated that the motor's winding limit was violated [52]. In particular, the average error between the commanded and measured torques was $6.03 \pm 3.62$ Nm with the linear implementation, compared to and $1.16 \pm 1.08$ Nm with the torsional implementation. For the damping level of 0.3 Nms/°, the linear implementation was unstable, while the torsional implementation was stable with an average torque-tracking error of $2.39 \pm 1.89$ Nm.

## 6.3. Stair Ascent with an above-Knee Amputee

To demonstrate the behavior of the torque-sensitive transmission in activities involving high levels of torque at high output angles, we carried out pilot tests with an above-knee amputee subject performing stair ascent using both implementations of the torque-sensitive transmission. The amputee participant (30 years old, 65 kg body mass, Figure 8a) provided written consent to the University of Utah's IRB-approved clinical testing protocols. For data collection, the participant was fitted to a bionic leg prosthesis by a certified prosthetist. The bionic leg comprised a powered knee prosthesis used in this study, and a powered ankle prosthesis from [48]. For each implementation, the participant was asked to perform the stair ascent movement five times, using a controller that provides high levels of torque at high output knee angles for a timely vertical propulsion of the body's center of mass [53].

The participant was successful in climbing stairs using both implementations of the torque-sensitive transmission, and the provided torque profiles were similar (Figure 8b). However, in the torsional implementation, the torque-sensitive joint started moving sooner, and reached its maximum position at $32.3 \pm 3.6\%$ of the movement, compared to $46.7 \pm 5.4\%$ with the linear implementation (Figure 8c). This resulted in a timelier modulation of the torque ratio for the torsional implementation, and correspondingly better motor performance. In particular, the Joule heating loss of the motor was $32.1 \pm 5.6$ J with the torsional implementation, 18% lower than $39.4 \pm 5.1$ J with the linear implementation (Figure 8d). In addition, peak current of the motor was $18.12 \pm 3.2$ A with the torsional implementation, 17% lower than $22.0 \pm 2.7$ A with the linear implementation (Figure 8e).

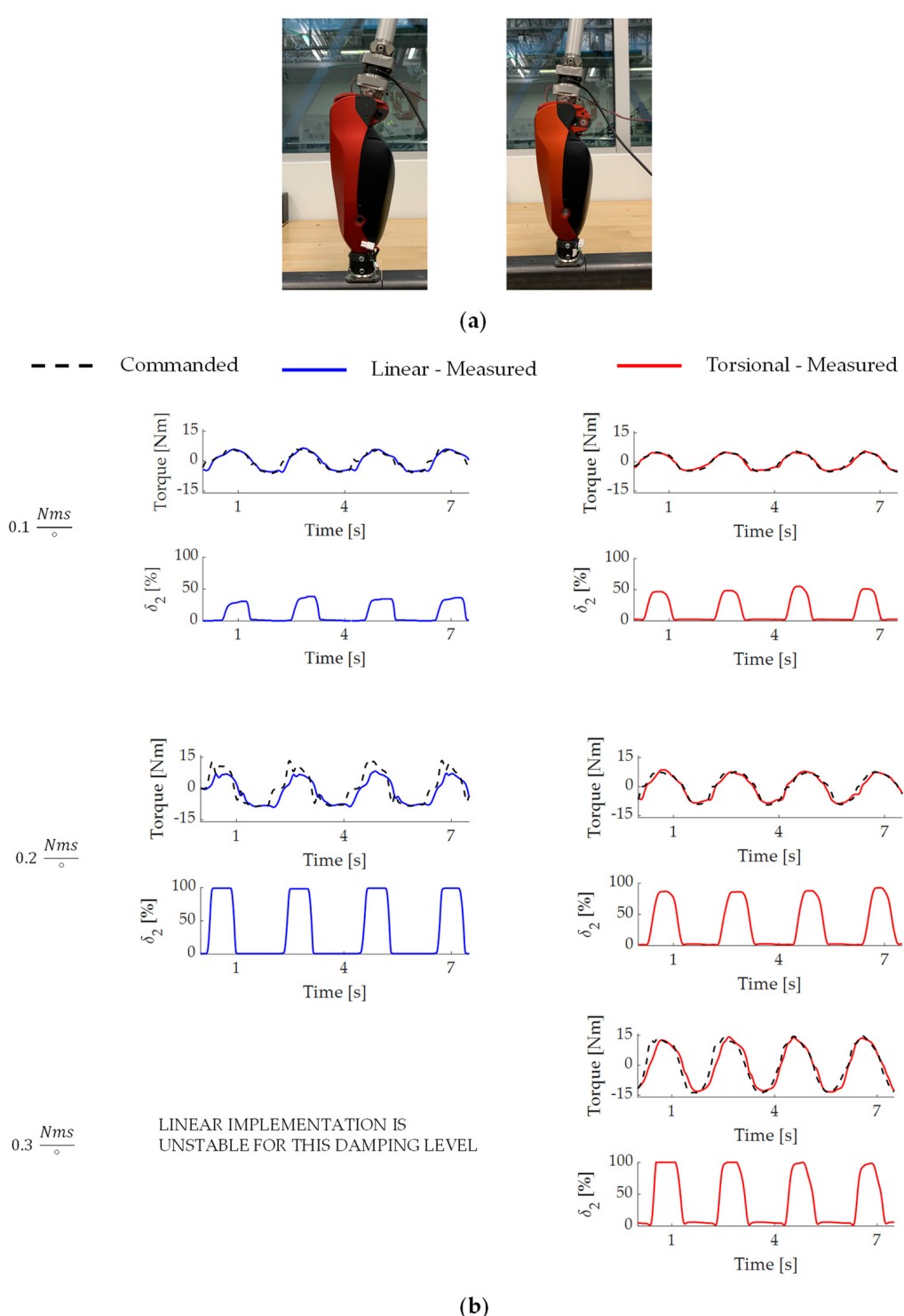

**Figure 7.** Damping emulation on the bench. (**a**) Experimental setup. A participant used a pylon to manually back-drive the knee joint, which was programmed to act as a damper. (**b**) Commanded and measured output torque, along with movement of the torque-sensitive joint under three different commanded damping conditions (0.1, 0.2, and 0.3 Nms/°). Results are shown for both implementations.

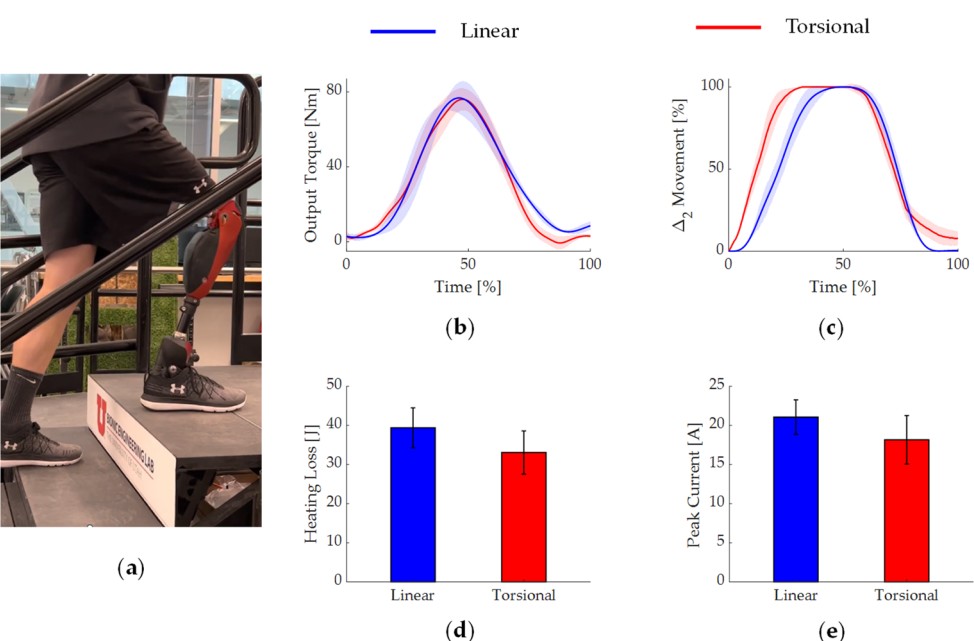

**Figure 8.** Stair ascent with an above-knee amputee participant. (**a**) A snapshot of the participant ascending the staircase. (**b**) Output torque profile at the knee joint during the assistive phase. (**c**) Movement of the torque-sensitive joint. (**d**) Heating losses at the motor. (**e**) Peak electrical current required at the motor. Results are shown for both implementations.

## 7. Discussion

Torque-sensitive or passively variable transmission systems present a promising solution to enhance the torque and power density of wearables and other robotic devices [44,46,48], bringing them closer to having human-like performance and achieving real-world impact. In this paper, we propose using the sensitivity between the input displacement and the output torque as a key metric to assess important outcomes of these advanced mechanisms, such as controllability and efficiency. To this end, we calculated and analyzed the sensitivity values of two different torque-sensitive transmissions across their workspace using analytical models. Our experimental results with these two designs implemented in a knee prosthesis confirmed the accuracy of the sensitivity model (Figure 6c). Moreover, these results show that the analysis of sensitivity reveals important characteristics of torque-sensitive and passively variable transmissions.

Our study uses two different implementations of a torque-sensitive transmission as a tool to demonstrate the connection between the sensitivity of a specific design and its performance as a robot actuator. With this goal in mind, we selected the two implementations of the torque-sensitive transmission to avoid factors that might confound the relationship between sensitivity and performance. Specifically, the two implementations have similar overall transmission ratios as well as similar trends in how the transmission ratio changes as a function of the output joint position (Figure 2e). Moreover, the two implementations show similar changes in transmission ratio when an output torque is applied because the torque-sensitive joints have similar travel range (Figure 2e). Both implementations are housed in the same powered knee prosthesis frame and powered by the same motor and linear drive. Finally, we provided torque to the knee prosthesis using an open-loop controller, which achieved the same level of torque-tracking ability in both implementations (Figure 6a). Thus, the experimental comparisons between the two implementations of the torque-sensitive actuator are reflecting differences that are primarily the result of a different sensitivity.

Analysis carried out using the analytical model suggests that, compared to the linear implementation, the torsional implementation has a more uniform sensitivity across the workspace (Figure 5e,f). In particular, compared to the linear implementation, the torsional

implementation has lower absolute sensitivity (i.e., relative sensitivity <1) for small angles of the output joint (0–30°) but higher absolute sensitivity (i.e., relative sensitivity >1) at larger angles of the output joint (>60°) (Figure 5e–g). Experiments confirm that this fundamental difference in sensitivity distribution is directly related to controllability and efficiency outcomes, as outlined in Section 3.

During damping emulation experiments on the bench under low output angles, the torsional implementation was able to stably provide damping resistance for all three commanded damping levels. The linear implementation, on the other hand, showed signs of voltage saturation starting with the medium damping level of 0.2 Nms/°, and became unstable at the highest damping level of 0.3 Nms/° (Figure 7b). These results are in agreement with the expected outcomes from the sensitivity analysis, which suggested that the lower sensitivity of the torsional implementation would lead to better controllability and protection against voltage saturation. Consequently, one would expect the linear implementation to outperform the torsional implementation in damping emulation for larger angles. However, at that range of output angle, the absolute sensitivity of both implementations is much lower than that of the output angles used in the benchtop experiment (i.e., 5–30°). We thus speculate that the linear implementation would only show better controllability with damping values larger than 0.3 Nms/°. To the best of our knowledge, such a level of damping is not used in powered prosthesis controllers [49,50], but might be beneficial in other robotic applications.

When the amputee subject climbed stairs with the prosthesis, the torque-sensitive joint in the torsional implementation reached its maximum position sooner than in the linear implementation. These results are expected from the analysis, given the generally higher sensitivity of the torsional implementation. In addition, the analysis conveys that the output angle at which the calculated sensitivity drops to zero is larger for the torsional implementation. This analysis means that the torque-sensitive joint is expected to move earlier during stair ascent under the presence of output torque, and this phenomenon was also observed in experimental data (Figure 8). The earlier beginning and completion of the torque-sensitive movement led to a timelier increase of torque ratio, and consequently to a more efficient operation of the motor and lower peak loads on the main components of the drivetrain. At lower output angles, one might expect the linear implementation to have a more efficient motor operation under high-torque tasks, due to its lower sensitivity (Figure 5g). However, benchtop experiments showed that there is no significant benefit from exceeding a certain level of sensitivity. Moreover, the damping emulation experiment also showed that the linear implementation can experience voltage saturation at the same range of the output angle, a negative consequence of high sensitivity.

From the analysis and experimental results, we can conclude that rather than aiming for minimal or maximal sensitivity, an appropriate level for the expected loads and operating conditions should be the goal of the designer. For each area of interest within the actuator workspace, the sensitivity should not cross an upper threshold that would cause the torque-sensitive joint to move too quickly, causing voltage saturation and instability. At the same time, the sensitivity should not go below a certain limit that would prevent a timely modulation of the torque ratio under the expected loads. Notably, both the upper and lower thresholds directly depend on the output torque–speed requirements. Therefore, a general understanding of the tasks that the torque-sensitive actuator needs to perform is necessary to determine the target sensitivity. In this study, the torsional implementation outperformed the linear implementation in both experiments (i.e., damping emulation and stair ascent), but it is not objectively a better design and might underperform if the torque, position, or speed requirements at the output were different. Another important factor to consider is that the target level of sensitivity depends on the linear actuator design, as two actuators with similar sensitivity profiles might perform differently with different motors and linear drives. Therefore, torque-sensitive transmissions cannot be designed in isolation from the other components of the actuation system.

Both benchtop and amputee experiments provided consistent evidence for the relationship between sensitivity and actuator performance, proving the potential of this metric in quantifying and comparing the overall dynamic behaviors of different actuator designs. However, we also believe that sensitivity alone is not sufficient for designers to create suitable robots for the intended applications. In the future, we will focus on the development of a unified design framework that combines sensitivity analysis with other tools such as dynamic range analysis [44] as well as task-based and transmission-based simulations [42,48] and controller-specific considerations. All these metrics combined are necessary to optimize meaningful output variables such as peak and root-mean-square of motor current, battery consumption, and loads on main transmission components. Additionally, while the input displacement–output torque sensitivity proposed in this manuscript has merits, we will also formulate, analyze, and experimentally verify other sensitivity metrics related to different key variables (e.g., input torque, output displacement, or torque-sensitive joint displacement) to obtain a more complete analysis of torque-sensitive and passively variable transmissions.

## 8. Conclusions

In this paper, we introduced a measure of sensitivity of torque-sensitive and passively variable transmissions based on input displacement and output torque and investigated the use of that sensitivity as a key metric for the analysis of actuator performance. The experimentally validated model of sensitivity provided meaningful insights into the dynamic performance of different designs across their large workspace and, thus, can provide a meaningful tool for robot designers to better analyze these complex transmission systems. Future work will focus on the systematic analysis of other potential sensitivity metrics and the development of a unified design framework in which sensitivity analysis complements other analysis routines.

## 9. Patents

Tommaso Lenzi and Minh Tran are co-inventors of the torque-sensitive actuator used in this study (US Patent App. 17/269,627).

**Author Contributions:** Conceptualization, M.T. and T.L.; methodology, M.T., L.G. and T.L.; software, M.T., L.G. and T.L.; validation, M.T., L.G. and T.L.; formal analysis, M.T., L.G. and T.L.; writing—original draft preparation, M.T. and T.L.; writing—review and editing, M.T., L.G. and T.L.; funding acquisition, T.L. All authors have read and agreed to the published version of the manuscript.

**Funding:** This research was funded by the National Institutes of Health, grant number R01HD098154, the Department of Defense, grant number W81XWH2110037, and the Rocky Mountain Center for Occupational and Environmental Health through NIOSH ERC grant number T420H008414.

**Data Availability Statement:** All data needed to support the conclusions of this manuscript are included in the main text.

**Acknowledgments:** We thank Stein Witt for his help with the benchtop experiments. We also thank the amputee subject for his participation in the study.

**Conflicts of Interest:** The patent of the torque-sensitive actuator used in this study has been licensed to Ottobock SE & Co. KGaA. The funders had no role in the design of the study; in the collection, analyses, or interpretation of data; in the writing of the manuscript; or in the decision to publish the results.

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
