# Peer review of "Analysis and Validation of Sensitivity in Torque-Sensitive Actuators"

_actuators, doi:10.3390/act12020080_

Round 1

Reviewer 1 Report

The authors present a sensitivity modeling and analysis of passively variable transmission actuators to enhance their capabilities in terms of torque and power, making closer to human-like behavior.

This works is well written, clear and technically sound. The model was validated experimentally showing the improve in efficiency of this king of mechanisms. This is of great interest for the robotics community.

Author Response

We would like to sincrely thank the reviwer for their support of our work!

Reviewer 2 Report

The presents the sensitivity between input displacement and output torque as a new key metric to analyze the capability of passively variable transmissions and torque-sensitive transmissions to perform dynamic tasks. The linkage-based torque-sensitivity transmission system was used as the analytic target, and a real knee prosthesis was implemented to valid the sensitivity models. The paper is well written, described and organized.  

The questions and comments are following:

1.      In the Introduction section, the authors state that no analysis has been done regarding the dynamic interactions among the input joint, output joint and elastic torque-sensitive element during operation. The reviewer would like to know whether considering these dynamic interactions could avoid the motor saturation? How to perform your analysis to the real application? It is better to add the real approach, e.g., control method, to enhance your motivation.

2.     In Section 6.1, it is difficult to understand the open-loop torque controller. Does it mean that a current or torque signal was sent to the motor controller hardware? Please clarify the items.

3.     Please double check the symbols in the equations, e.g., atan of Equation 8 should not be italic.  

Author Response

We would like to thank the reviewer for their constructive feedback. We have provided a point-by-point response to their comments below, and have also modified the manuscript per their suggestions.

1)

We agree that the selection and implementation of different control methods would have a great impact on the performance of torque-sensitive actuators.

In this paper, we chose to use an open-loop torque controller, meaning that the commanded joint torque is converted to a commanded motor current through a model of the transmission system. This open-loop strategy is very common in powered prosthetics, as eliminates the need for torque-sensing components and simplifies the system hardware.

Using this open-loop controller, we predicted through simulation and experimentally verified in this manuscript that motor saturation during dynamic tasks such as impedance rendering can be avoided by designing an actuator with lower sensitivity values (i.e. the torsional implementation). However, the same actuator design might still suffer from motor saturation using a different control strategy (e.g. closed-loop torque control).

In the manuscript, we have highlighted the lack of controller considerations as a limitation for future work:

“However, we also believe that sensitivity alone is not sufficient for designers to create suitable robots for the intended applications. In the future, we will focus on the development of a unified design framework that combines sensitivity analysis with other tools such as dynamic range analysis [44], as well as task-based and transmission-based simulations [42], [48], and controller-specific considerations.”

2)

We apologize for the lack of information on controls strategy. Similar to point 1, we have included in the manuscript a description of the controller:

“In this paper, we chose to use an open-loop torque controller, meaning that the commanded joint torque is converted to a commanded motor current through a model of the transmission system. This open-loop strategy is very common in powered prosthetics, as eliminates the need for torque-sensing components and simplifies the system hardware.”

3)

We have done another check of the modeling section and corrected inconsistencies in the notations.